# Non-Overt Coagulopathy in Non-ICU Patients with Mild to Moderate COVID-19 Pneumonia

**DOI:** 10.3390/jcm9061781

**Published:** 2020-06-08

**Authors:** Daniela Mazzaccaro, Francesca Giacomazzi, Matteo Giannetta, Alberto Varriale, Rosa Scaramuzzo, Alfredo Modafferi, Giovanni Malacrida, Paolo Righini, Massimiliano M. Marrocco-Trischitta, Giovanni Nano

**Affiliations:** 1Operative Unit of Vascular Surgery, IRCCS Policlinico San Donato, San Donato Milanese, 20026 Milan, Italy; matteogiannetta@hotmail.it (M.G.); alberto.varriale@unimi.it (A.V.); alfredomodafferi@hotmail.it (A.M.); gmalacrida@libero.it (G.M.); paolo.righini@grupposandonato.it (P.R.); massimiliano.marroccotrischitta@grupposandonato.it (M.M.M.-T.); giovanni.nano@libero.it (G.N.); 2Cardiovascular Department, IRCCS Policlinico San Donato, San Donato Milanese, 20026 Milan, Italy; francesca.giacomazzi@grupposandonato.it; 3Operative Unit of University General Surgery, IRCCS Policlinico San Donato, San Donato Milanese, 20026 Milan, Italy; rosa.scaramuzzo@grupposandonato.it; 4Department of Biomedical Sciences for Health, University of Milan, 20124 Milan, Italy

**Keywords:** disseminated intravascular coagulation, DIC, COVID-19, respiratory distress

## Abstract

Introduction: Aim of the study is to assess the occurrence of early stage coagulopathy and disseminated intravascular coagulation (DIC) in patients with mild to moderate respiratory distress secondary to SARS-CoV-2 infection. Materials and methods: Data of patients hospitalized from 18 March 2020 to 20 April 2020 were retrospectively reviewed. Two scores for the screening of coagulopathy (SIC and non-overt DIC scores) were calculated. The occurrence of thrombotic complication, death, and worsening respiratory function requiring non-invasive ventilation (NIV) or admission to ICU were recorded, and these outcomes were correlated with the results of each score. Chi-square test, receiver-operating characteristic curve, and logistic regression analysis were used as appropriate. *p* Values < 0.05 were considered statistically significant. Results: Data of 32 patients were analyzed. Overt-DIC was diagnosed in two patients (6.2%), while 26 (81.2%) met the criteria for non-overt DIC. Non-overt DIC score values ≥4 significantly correlated with the need of NIV/ICU (*p* = 0.02) and with the occurrence of thrombotic complications (*p* = 0.04). A score ≥4 was the optimal cut-off value, performing better than SIC score (*p* = 0.0018). Values ≥4 in patients with thrombotic complications were predictive of death (*p* = 0.03). Conclusions: Overt DIC occurred in 6.2% of non-ICU patients hospitalized for a mild to moderate COVID-19 respiratory distress, while 81.2% fulfilled the criteria for non-overt DIC. The non-overt DIC score performed better than the SIC score in predicting the need of NIV/ICU and the occurrence of thrombotic complications, as well as in predicting mortality in patients with thrombotic complications, with a score ≥4 being detected as the optimal cut-off.

## 1. Introduction

Unlike other SARS viruses, the SARS-CoV-2 is associated with coagulation abnormalities with a pro-coagulant pattern that may lead to thrombotic or hemorrhagic complications and organ failure in a notable proportion of COVID-19 patients admitted to intensive care unit (ICU) for severe disease [1].

In particular, a large percentage of patients affected by severe COVID-19 pneumonia who did not survive may meet the criteria for disseminated intravascular coagulation (DIC) [2].

Data on the possible existence of an early stage of DIC in COVID-19 patients with mild to moderate disease are lacking. In this setting of patients, typical coagulation abnormalities may arise after a non-overt phase, which is not easily recognizable until complications occur. 

Indeed, a mild coagulopathy could be likely induced by sepsis, since many of these patients meet the Third International Consensus Definitions for Sepsis (Sepsis-3) [3], even when the disease is not severe.

The International Society of Thrombosis and Hemostasis (ISTH) has provided an interim guidance for risk stratification of coagulopathy at admission in COVID-19 patients [4]; however, even if some scoring systems have been proposed, screening and diagnosis of overt and non-overt DIC still represent a challenge [5,6].

Aim of the study is to report the occurrence of early stage coagulopathy and disseminated intravascular coagulation (DIC) in patients with mild to moderate respiratory distress secondary to SARS-CoV-2 infection, admitted to a non-ICU ward of a University Hospital in the Milan metropolitan area. The sepsis-induced coagulopathy (SIC) score, the overt-DIC score and the non-overt DIC score were utilized, according to the algorithm proposed by the International Society of Thrombosis and Hemostasis (ISTH). Furthermore, the prognostic value of SIC score and of non-overt DIC score for the prediction of mortality and complications was assessed.

## 2. Materials and Methods

The Ethics Committee of IRCCS Ospedale San Raffaele approved this retrospective study on May 5 2020. Specific informed consent was waived.

Clinical, laboratory, and radiologic data of patients admitted to our ward for mild to moderate COVID-19 respiratory distress from 18 March 2020 to 20 April 2020 were retrospectively reviewed searching in the electronical internal database. 

Patients’ demographics were recorded (age, sex, and body-mass index-BMI), as well as the presence of comorbidities such as diabetes mellitus (defined as fasting glucose levels >100 mg/dL or the intake of at least one drug to control the serum glucose levels), arterial hypertension (defined as systolic blood pressure >140 mmHg and/or diastolic blood pressure >90 mmHg, or as taking at least one drug for blood pressure control), smoking habits (defined as current or past smoker versus non-smoker), chronic obstructive pulmonary disease (COPD), coronary artery disease, chronic renal failure (defined as an estimated Glomerular Filtration Rate-GFR < 60 mL/min/1.73 m^2^), and history of previous or active neoplasm.

The time elapsing between the onset of symptoms and hospital admission, as well as the number of days with fever (i.e., axillary cutaneous temperature greater than 37.5 °C) were recorded as well. Furthermore, the arterial partial pressure of oxygen/fraction of inspired oxygen (PaO2/FIO2) ratio was recorded at presentation and during hospital stay.

Laboratory data on admission included white blood cells count with neutrophil/lymphocyte ratio (NLR), hemoglobin and red blood cells count, platelets count, prothrombin time (PT), activated partial thromboplastin time (aPTT), D-dimer, fibrinogen, inflammatory markers (C-Reactive Protein-CRP, ferritine), renal and hepatic function, high sensitivity (hs)-troponin and interleukin-6. During the hospital stay, laboratory tests were repeated with a variable frequency according to the patients’ clinical course (daily, every other day, or twice a week).

The sequential organ failure assessment (SOFA) score was calculated in every patient to verify or rule out the presence of septic status, according to the new Sepsis-3 definition [3]. The SOFA score assesses organ dysfunction (respiratory system, nervous system, cardiovascular system, liver function, coagulation, and renal function) and quantifies abnormalities according to clinical findings, laboratory data, or therapeutic interventions (Table 1). A SOFA score ≥2 consequent to infection defines the presence of septic status [3].

The diagnosis of COVID-19 pneumonia was confirmed by real-time reverse-transcription polymerase chain reaction on nasopharyngeal swab and chest X-ray performed on admission in the emergency department. 

All patients according to internal protocol underwent a computed tomography pulmonary angiography (CTPA) to assess the pulmonary parenchyma and the possible occurrence of pulmonary vessels thrombosis (PVT). Furthermore, all patients underwent a Duplex scan of the veins and arteries of the upper and lower limbs to investigate the presence of peripheral thrombosis.

All patients received anticoagulation treatment with low molecular weight heparin (LMWH) at prophylactic dosage on admission, except those who were already taking an oral anticoagulant for other medical conditions. When PVT or peripheral thrombosis were found, LMWH therapy was increased to anticoagulation dosage (i.e., 100 international unit/kg of enoxaparin twice a day, adjusted for glomerular filtration rate). 

The assessment of coagulopathy was performed using the diagnostic algorithm established by the ISTH [5], for each day of hospitalization in which the complete blood tests were available. The highest value of the scores among those recorded was then retained. 

The SIC scoring system is based on the use of two laboratory parameters (platelets count and international normalized ratio of PT) in addition to the SOFA score [5] (Figure 1). According to the ISTH’s algorithm, the SIC score in septic patients should be calculated first, and if the SIC score is ≥4 points, then the overt-DIC score should be applied. 

The non-overt DIC score is based on major criteria that encompasses quantitative measurement of the platelets, the PT and the fibrinogen degradation products, but also emphasizes the trend of the above parameters over time. In addition, some specific criteria (such as antithrombin levels or protein C levels) can further be used for a better refining of the risk assessment [6] (Figure 1).

SIC score was therefore calculated first. If the SIC score was ≥4, the overt-DIC score was calculated. In particular for DIC score, D-dimer levels ≥1 µg/mL scored 2 and ≥4 µg/mL scored 3, as recommended at the 49th Scientific Subcommittee meeting of the ISTH. Patients were therefore classified has having DIC if overt-DIC score was ≥5. If overt-DIC score was <5, or SIC score was <4, the scoring system to diagnose non-overt DIC was further calculated (Figure 2), considering only major criteria. 

Primary outcomes were the occurrence of thrombotic complication, death, and worsening respiratory function requiring non-invasive ventilation (NIV) or admission to ICU during the hospital stay. These outcomes were correlated to the results of the SIC score, the overt DIC score, and non-overt DIC score.

Statistical analysis was performed using STATA 16.1 (StataCorp LLC, College Station, TX, USA). Continuous variables were reported as mean ± standard deviation (SD) for normally distributed data, otherwise median and interquartile range (IQR) were reported. The normality of the continuous variables was checked using the Shapiro-Wilk test. Categorical variables were presented as *n* (%). Chi-Square test and logistic regression analysis were used as appropriate to evaluate the association between variables. Receiver-operating characteristic (ROC) analysis was used to determine the prognostic value of the scores in predicting death, the occurrence of thrombotic complications, or the worsening of respiratory function requiring NIV or admission to ICU.

An alpha value of 0.05 was the level reference set for statistical significance.

## 3. Results

Data of 32 patients who were admitted in our non-ICU ward for a mild to moderate COVID-19 respiratory distress were analyzed. Of them, 23 (71.9%) were male. Patients’ mean age was 68.6 ± 12 years. As described in Table 2, most of them were affected by systemic hypertension (20 patients, 62.5%) and had a history of previous smoking (14 patients, 43.7%).

On admission, patients presented in about half of cases with a pneumonia with mild respiratory distress (18 patients, 56.2%), while in the remaining cases a moderate respiratory distress syndrome was already present according to Berlin criteria [7]. Mean PaO2/FIO2 ratio on admission was 239.4 ± 100. The mean SOFA score was 2.8 ± 0.4.

Before hospital admission, most patient had had symptoms of dyspnea and fever for a mean of 7.2 ± 3.9 days. Fever, which was present in all patients, had a mean duration of 10.7 ± 4.9 days.

During the clinical course, 18 patients required NIV (56.2%) for worsening respiratory function, and in two of them admission to ICU was necessary (6.2%).

The CTPA revealed the presence of PVT in 21 cases (65.6%). In all these patients, the dosage of the LMWH was adjusted to full anticoagulation (enoxaparin 100 IU/kg twice a day adjusted for GFR).

Moreover, at duplex scan, a deep venous thrombosis was detected in one patient who had PVT (left popliteal vein), while a superficial thrombosis of varicose veins of the left thigh occurred in one patient who did not have PVT. No arterial thromboses were detected.

PVT was mainly located in the segmental (8 cases, 38.1%) and subsegmental arteries (7 cases, 33.3%), while in 5 cases it was detected at a lobar artery and in the remaining case at the main trunk. 

Three patients died (9.4%).

## 4. SIC Score

The SIC score was ≤3 in all patients except in two who had a score ≥4; in this cases the overt-DIC score was calculated. In the remaining patients, the non-overt DIC score was adopted.

## 5. Overt-DIC

Both patients with a SIC score ≥4 were confirmed to have overt DIC (6.2%), according to the ISTH’s criteria. One of these patients (male, 86 years old) had subsegmental PVT in more than two segments and was fully anticoagulated with LMWH (enoxaparin 6000 IU twice a day). Moreover, because of worsening respiratory function, NIV continuous positive airway pressure (cPAP) with helmet was necessary. Thereafter, patients developed an ischemic stroke and an ileo-psoas hematoma due to retroperitoneal bleeding which was treated with endovascular coils embolization of L2–L3 artery. Anticoagulation was therefore interrupted but unfortunately the patient died for multiorgan failure on day 15. 

In the other patient (female, 69 years old), a subsegmental PVT in more than two segments was detected at CTPA, despite the patient was already taking a novel oral anticoagulant for atrial fibrillation before admission. However, her respiratory function progressively worsened, requiring first NIV cPAP with helmet, and then admission to ICU for mechanical ventilation (Table 3).

In both cases, the highest score for overt DIC was recorded in the same day in which the CRP and the D-dimer levels were the highest, 1 day after fever disappearance. 

## 6. Non-Overt DIC

In the remaining 30 patients, the SIC score was lower than 4, therefore the non-overt DIC score was calculated, according to the proposed algorithm.

Four patients (12.5%) scored 0 at the non-overt DIC score. PVT was detected in two of these cases. No other thrombotic complications were recorded. Among these four patients, two required NIV (one of them had PVT and the other one did not have PVT at CTPA). None of them died or required admission to ICU (Table 3). 

In the other 26 patients (81.2%), the non-overt DIC score was not zero. Seventeen of these patients had PVT (65.4%). One case, (male, 83 years old patient) had a concurrent left popliteal vein thrombosis. His respiratory function worsened to require NIV, but he eventually died for severe respiratory failure on day 22. In the remaining 16 cases with PVT, NIV was required in 10 more patients. Among these 10, one patient needed admission to ICU, while another patient (female, 85 years old) died on day 27 for sepsis.

Among the 9 patients who did not have PVT at CTPA, one developed a superficial venous thrombosis of the varicose veins in the thigh, and 3 required NIV. 

Five patients neither showed any thrombotic complications or the need for NIV/ICU (Table 3).

## 7. Prognostic Value of SIC and Non-Overt DIC Scores

The values of SIC score ≥4 were not predictive of the need of NIV/ICU (*p* = 0.85), neither of the occurrence of death (*p* = 0.52) nor of thrombotic complications (*p* = 0.36).

According to ROC analyses, the area under curve (AUC) for SIC score when predicting the need for NIV/ICU was 0.54; when predicting mortality and the occurrence of thrombotic complications AUC was respectively 0.70 and 0.67. Sensitivity (SE), specificity (SP), positive predictive value (PPV), and negative predictive value (NPV) are described in Table 4.

On the other hand, when the non-overt DIC score was applied to all patients (including those who had an overt DIC), the values were significantly predictive of the need of NIV/ICU (*p* = 0.02) and of occurrence of thrombotic complications (*p* = 0.04) but not of death (*p* = 0.19).

In particular, according to ROC analyses, a score ≥4 was the optimal cut-off value to predict worsening respiratory function requiring NIV/ICU, performing better than SIC score (AUC = 0.81, *p* = 0.0018, Figure 3), with a sensitivity of 83.3% ± 17.2% and a specificity of 71.4% ± 23.7% (Table 4). The risk increase was 0.56 folds per unit.

A score ≥4 was also the optimal cut-off value to predict the occurrence of thrombotic complications (area under curve = 0.73, *p* = 0.04), with a sensitivity of 69.6% ± 18.8% and a specificity of 66.7% ± 30.8% (Table 4). The risk increase in this case was 0.64 folds per unit.

Values ≥4 in patients with thrombotic complications were highly predictive of death (*p* = 0.03, Table 4.)

## 8. Discussion

COVID-19 is a recently emerged systemic disease due to Sars-CoV-2 infection, with a significant impact on hemostasis, coagulation, and inflammatory system. From the beginning of the pandemic, several medical reports highlighted the raised thromboembolic risk of this disease, which could at least in part give reason for the referred high mortality rate and for those cases of unexplained sudden death [8]. All kind of venous and arterial thromboembolic complications were described [9], including ischemic stroke, acute coronary syndrome, arterial occlusion, vein thrombosis, and pulmonary embolism. 

Owing to the high risk for thromboembolic events in COVID-19 patients, coagulation screening testing including D-dimer, PT, aPTT, fibrinogen levels, and platelets count is strongly recommended on admission and during hospital stay [10].

Alterations of the blood coagulation parameters are a common finding among COVID-19 patients. Elevated D-dimer levels are consistently reported, whereas their gradual increase during the disease course has been associated with clinical worsening [11]. Furthermore, severe thrombocytopenia, elongation of PT, and of aPTT are typical of the severe stage of the disease and have been associated with increased mortality in critically ill ICU patients, irrespectively of the presence of ARDS [12].

All these parameters are suggestive of life-threatening DIC, which is likely triggered by several pathogenetic mechanisms, including both microangiopathy and macroangiopathy. Magro et al. [13] reported that microvascular injury could be mediated by activation of complement pathways. Other important issues include the increase of Von Willebrand factor (vWF) activity, of vWF antigen, of factor VIII, and presence of Lupus Anticoagulant, which is a probable consequence of endothelial inflammation and disruption, as reported by Helms et al. [1].

Furthermore, several reports of pulmonary emboli may indicate the presence of macrovascular injury, despite the presence or not of DIC [14]. Finally, bacterial superinfection that may develop over time in more severe cases can lead to further consumptive coagulopathy.

It is not yet clear whether the overt DIC described in seriously ill patients is preceded by a less evident phase of non-overt coagulopathy.

If so, screening for non-overt DIC in patients with mild to moderate disease could be fundamental to optimize the treatment and prevent the organ failure typical of the most severe stage of the disease. 

The diagnosis of non-overt DIC nevertheless is not straightforward. In COVID-19 patients, it could be even more challenging, given the possible alterations of the laboratory blood parameters due to the concomitant hyperinflammatory status and the interactions with anti-viral and immunosuppressive drugs that are usually used at early stage of the disease. The organ dysfunction in these cases may not be diagnosed, leading to the underestimation of a more complex clinical picture. 

Some scoring systems have been proposed by the ISTH for the screening and the diagnosis of DIC in its early stage, in patients without COVID-19, in particular the scoring system of the sepsis-induced coagulopathy (SIC) [5] and that of the non-overt DIC [6].

The rationale for the use of the SIC score lies in the fact that most patients with COVID-19 pneumonia fulfill the criteria for sepsis, according to the new Sepsis-3 definition [3], given the presence of the viral infection and the respiratory dysfunction. In our case series, in fact, the SOFA score was ≥2 in all cases, even if patients had a mild to moderate disease.

The SIC score well correlates with mortality [5], however its use has still not been validated. Moreover, its applicability in COVID-19 patients has not yet been described in mild to moderate disease.

In our case series, the SIC score resulted to be more specific than the non-overt DIC score when predicting the occurrence of thrombotic complications, the need for NIV/ICU, and mortality. Nevertheless, its sensitivity was much lower, therefore the SIC score had a negligible value when used for the stratification of the patients with a mild to moderate disease. 

On the other side, the non-overt DIC score proved to be a reliable prognostic tool for the need of NIV/ICU and the occurrence of thrombotic complications, even if only major criteria were used. Furthermore, when the non-overt DIC score was applied in patients who had thrombotic complications, detected either at the CTPA or at duplex scan, a score ≥4 was predictive of mortality. 

Therefore, according to our results, the non-overt DIC score proved to be superior to the SIC score for the screening of the possible presence of an activated but still compensated hemostatic system in patients affected by COVID-19 pneumonia with mild to moderate disease.

Since the specific criteria of the non-overt DIC score require the use of laboratory tests (antithrombin or protein C) that are not easily available in the emergency setting and that are typically not considered in routine clinical practice, concerns may arise about the accuracy of the score when using only major criteria [15], or when laboratory tests are not performed every day. However, beside the mere calculation of the score, the overall clinical evaluation of the patient’s status cannot be omitted, bearing in mind that the SARS-CoV-2 infection can be a multisystemic disease which may require daily or at least every other day assessment of laboratory tests, for a prompt and early recognition of possible organ failure. 

Our study has some limitations that could prevent from generalizing the results observed, being the most important the retrospective design of the study and the small sample size.

Further studies are needed to give insights to the correct recognition of a compensated imbalance of the hemostatic system in patients with mild to moderate COVID-19 pneumonia, to optimize risk stratification for a proper treatment.

## 9. Conclusions

Overt DIC occurred in 6.2% of the cases of non-ICU patients hospitalized for a mild to moderate COVID-19 pneumonia, while 81.2% fulfilled the criteria for a non-overt DIC. The non-overt DIC score performed better than the SIC score in predicting the need for NIV/ICU and the occurrence of thrombotic complications, as well as in predicting mortality in patients with thrombotic complications. In particular, a non-overt DIC score ≥4 was the optimal cut-off for predicting the need for NIV/ICU and the occurrence of thrombotic complications. If the occurrence of thrombosis was added to the score, a value ≥4 was also prognostic for mortality.

## Figures and Tables

**Figure 1 jcm-09-01781-f001:**
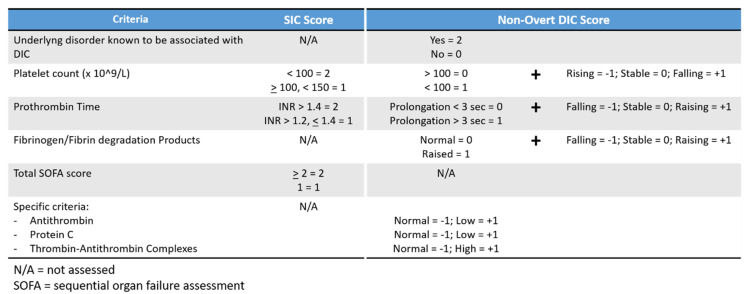
The screening of coagulopathy (SIC) and the non-overt disseminated intravascular coagulation (DIC) scoring systems.

**Figure 2 jcm-09-01781-f002:**
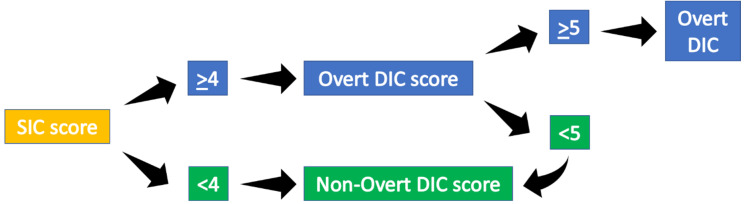
Algorithm that was used in our case series for the screening of overt and non-overt DIC, according to the recommendation of the International Society of Hemostasis and Thrombosis (ISTH).

**Figure 3 jcm-09-01781-f003:**
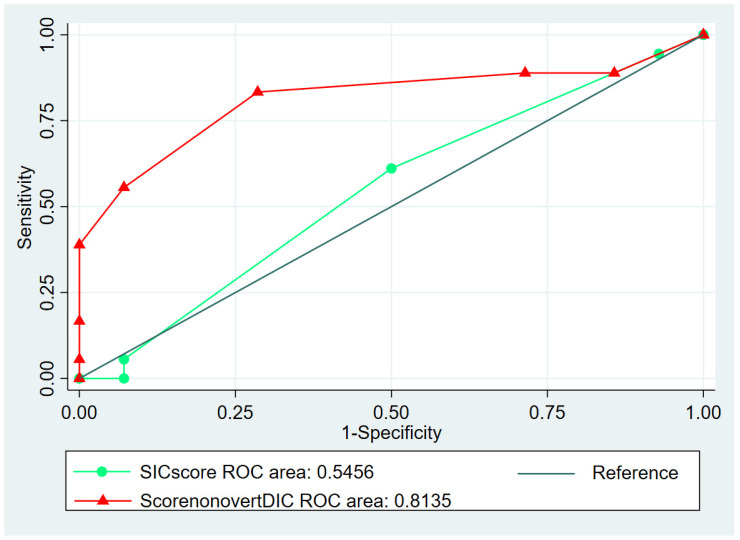
Receiver operating characteristic (ROC) analysis of the SIC score (curve with circles) and the non-overt DIC score (curve with the triangles) in predicting respiratory function requiring NIV/ICU. Area under curve is reported for both.

**Table 1 jcm-09-01781-t001:** Sequential organ failure assessment (SOFA) score.

	**SOFA Score**
**PaO2/FiO2 (mmHg)**	
≥400	0
<400	+ 1
<300	+ 2
<200 AND mechanically ventilated	+ 3
<100 AND mechanically ventilated	+ 4
**Glasgow Coma Scale**	
15	0
13–14	+ 1
10–12	+ 2
6–9	+ 3
<6	+ 4
**Mean Arterial Pressure or Administration of Vasopressors Required**	
MAP ≥ 70 mmHg	0
MAP < 70 mmHg	+ 1
Dopamine ≤ 5 µg/kg/min or dobutamine (any dose)	+ 2
Dopamine > 5 µg/kg/min OR epinephrine ≤ 0.1 µg/kg/min OR nor epinephrine ≤ 0.1 µg/kg/min	+ 3
Dopamine > 15 µg/kg/min OR epinephrine > 0.1 µg/kg/min OR nor epinephrine > 0.1 µg/kg/min	+ 4
**Bilirubin (mg/dL)**	
<1.2	0
1.2–1.9	+ 1
2.0–5.9	+ 2
6.0–11.9	+ 3
>12	+ 4
**Platelets × 10^3^ µL**	
≥150	0
<150	+ 1
<100	+ 2
<50	+ 3
<20	+ 4
**Creatinine (mg/dL)**	
<1.2	0
<1.2–1.9	+ 1
<2.0–3.4	+ 2
<3.5–4.9	+ 3
>5.0	+ 4

PaO2/FiO2 = arterial partial pressure of oxygen/fraction of inspired oxygen; MAP = mean arterial pressure.

**Table 2 jcm-09-01781-t002:** Demographics, clinical, and laboratory data of patients of the case series.

	**Data of Patients (*n* = 32)**
Age (mean ± SD)	68.6 + 12 years
Male sex (*n*, %)	23 (71.9%)
BMI	27.1 ± 4.3
Hypertension	20 (62.5%)
Dyslipidemia	5 (15.6%)
Diabetes Mellitus	7 (21.9%)
Coronary Artery Disease	7 (21.9%)
Chronic Obstructive Pulmonary Disease	1 (3.1%)
History of previous/current smoke	14 (43.7%)
PaO2/FiO2 on admission (mean ± SD)	239.4 ± 100.1
**Laboratory data**	
Hemoglobin (mean ± SD)	11.7 ± 2.2 g/dL
NLR (mean ± SD)	7.8 ± 7.7
Platelets (mean ± SD)	186.5 ± 83.4 * 10^3^ U/µL
PT (mean ± SD)	14.5 ± 3.9 sec.
INR (mean ± SD)	1.16 ± 0.27
aPTT (mean ± SD)	34.3 ± 4.7 sec.
D-dimer (mean ± SD)	3.7 ± 4.8 µg/mL
Fibrinogen (mean ± SD)	585.7 ± 135.7 mg/dL
LDH (mean ± SD)	582.1 ± 216.4 U/L
CRP-hs (mean ± SD)	9.1 ± 8.1 mg/dL
IL-6 (mean ± SD)	217.1 ± 355.1 pg/mL
Ferritin (mean ± SD)	1225.1 + 1120.1 µg/L
Hs-Troponin (mean ± SD)	20.4 + 0.3 ng/L
Creatinine (mean ± SD)	0.85 + 0.3 mg/dL
AST (mean ± SD)	42.3 + 2.8 U/L
ALT (mean ± SD)	51.3 + 3.9 U/L

SD = standard deviation. BMI = body mass index. PaO2/FiO2 = arterial partial pressure of oxygen/fraction of inspired oxygen. NLR = neutrophil lymphocyte ratio. PT = prothrombin time. INR = international normalized ratio. aPTT = activated partial thromboplastin time. LDH = lactate dehydrogenase. CRP = C-reactive protein. Hs = high sensitivity. IL6 = interleukin 6. AST = aspartate aminotransferase. ALT = alanine aminotransferase.

**Table 3 jcm-09-01781-t003:** Occurrence of death, thrombotic complications, and need for non-invasive ventilation (NIV) and intensive care unit (ICU) admission for the patients of the case series, according to the results of the overt and non-overt disseminated intravascular coagulation (DIC) scores.

	Overt DIC and Non-Overt DIC Scores	Complications
32 patients	2 patients with overt DIC score ≥5	1 patient: PVT + NIV (Death) 1 patient: PVT + NIV (ICU)
	4 patients with non-overt DIC score = 0	1 patient: PVT 1 patient: PVT + NIV 1 patient: NIV 1 patient: no complications
	26 patients with non-overt DIC score ≥1	17 patients: PVT - 11 NIV (2 Deaths, 1 ICU) 9 patients: no PVT - 1 Superficial Vein Thrombosis - 3 NIV - 5 no complications

PVT = pulmonary vessel thrombosis.

**Table 4 jcm-09-01781-t004:** Sensitivity (SE), specificity (SP), positive predictive value (PPV), and negative predicting value (NPV) of the SIC and of the non-overt DIC scores at cut-off values ≥4 for the need of non-invasive ventilation (NIV)/intensive care unit (ICU) admission and occurrence of thrombosis and mortality, with each respective *p* value. 95% confidence intervals are reported.

	SIC Score ≥ 4	*p* Value	Non-Overt DIC Score ≥ 4	*p* Value
**Need for NIV/ICU**	SE: 5.6% ± 10.6%	*p* = 0.85	SE: 83.3% ± 17.2%	***p* = 0.02**
	SP: 92.8% ± 13.5%		SP: 71.4% ± 23.7%	
	PPV: 50% ± 35.3%		PPV: 78.9% ± 9.3%	
	NPV: 43.3% ± 9%		NPV: 76.9% ± 11.7%	
**Thrombotic Complications**	SE: 0%	*p* = 0.36	SE: 69.6% ± 18.8%	***p* = 0.04**
	SP: 81.8% ± 22.8%		SP: 66.7% ± 30.8%	
	PPV: 0%		PPV: 84.2% ± 8.4%	
	NPV: 30% ± 8.4%		NPV: 46.1% ± 13.8%	
**Mortality ***	SE: 40% ± 42.9%	*p* = 0.52	SE: 100%	***p* = 0.03**
	SP: 100%		SP: 55.2% ± 18.1%	
	PPV: 100%		PPV: 18.7% ± 9.7%	
	NPV: 90% ± 5%		NPV: 100%	

* SE, SP, PPV, and NPV of the non-overt DIC score for mortality are referred to patients who had thrombotic complications. For the SIC score, the SE, SP, PPV, and NPV were the same either when referred to patients who had or when referred to patients who did not have thrombotic complications.

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
