# Peer review of "Non-Overt Coagulopathy in Non-ICU Patients with Mild to Moderate COVID-19 Pneumonia"

_jcm, 2020, doi:10.3390/jcm9061781_

Round 1

Reviewer 1 Report

Authors examined the occurrence of early stage coagulopathy and DIC in patients with mild and moderate respiratory distress secondary to SARS-CoV-2 infection admitted to a non-ICU ward. Authors also examined SIC, overt-DIC and non-overt DIC scores in these patients.

Although this manuscript is potentially interesting, several issues arise.

Major points

  • Study size was small.
  • It is interesting that these patients with SARS-CoV-2 infection compared without SARS-CoV-2 infection.
  • Non-overt-DIC score varied among various reports.
  • Why were there many PE patients (21/32)? Please discuss?

Were they treated with anticoagulant?

Minor points

  • What is NIV in Table 2?

Author Response

Response to Reviewer’s Comments #1

Dear Reviewer,

we are grateful for your preciuos time spent in the revision of our paper. Major and minor points have been addressed and changes have been made in the text. We believe that the paper has now been improved according to the Reviewer’s suggestion.

Hereafter You’ll find the detailed responses to your comments. Thanks again for your efforts.

Authors examined the occurrence of early stage coagulopathy and DIC in patients with mild and moderate respiratory distress secondary to SARS-CoV-2 infection admitted to a non-ICU ward. Authors also examined SIC, overt-DIC and non-overt DIC scores in these patients.

Although this manuscript is potentially interesting, several issues arise.

Major points

  1. Study size was small.

We agree with the Reviewer about this consideration but unfortunately in this retrospective analysis we had complete data only of this cohort of patients. We have aknowledged this as a limit of our paper, in the Discussion section (please see Discussion page 10 line 335).

  1. It is interesting that these patients with SARS-CoV-2 infection compared without SARS-CoV-2 infection.

The consideration of the Reviewer is interesting. Nevertheless, we excluded in the design of our study a possible comparison of the coagulopathy of COVID-19 patients with that of non-COVID-19 patients. The data currently available in the Literature highlight that coagulation parameters may be significant different between the two groups (Helms et al. Intensive Care Med 2020. https://doi.org/10.1007/s00134-020-06062-x), therefore we thought that this difference could affect the methodology of our study leading to an important bias.

  1. Non-overt-DIC score varied among various reports.

This is a crucial point and the Reviewer has perfectly focused the subject: that’s one of the reasons why the diagnosis of a possible non-overt DIC may be not straightforwarding.

  1. Why were there many PE patients (21/32)? Please discuss?

The finding of PE patients is comparable to the other reports currently available in the Literature (Helms et al. Intensive Care Med 2020. https://doi.org/10.1007/s00134-020-06082-7). Of course, the incidence of PE may be affected by the number of computed tomography pulmonary angiography (CTPA) that are performed. In our internal clinical protocol, all patients underwent a CTPA. Therefore, the incidence of PE is higher if compared to other reports as a consequence of our screening protocol.

  1. Were they treated with anticoagulant?

As described in the results and according to the international guidelines, all patients with PE were treated with anticoagulant (please see Results page 7, lines 192-193).

Minor points

  1. What is NIV in Table 2?

NIV stands for Non-Invasive Ventilation. A list of the abbreviation used has been added at the beginning of the manuscript on page 1 and 2 (lines 38-61). Furthermore, the abbreviation NIV was spelt out in the Table’s legend (please see page 7 line 232).

Reviewer 2 Report

The authors retrospectively reviewed their experience of non-overt coagulopathy in non-ICU patients with mild to moderate COVID-19 pneumonia to assess the occurrence of early coagulopathy and disseminated intravascular coagulation (DIC) in patients with mild to moderate respiratory distress from 18/03/2020 to 20/04/2020. Were retrospectively reviewed. They calculated both SIC and non-overt DIC scores and examined thrombotic complications, death, and worsening respiratory function requiring non-invasive ventilation (NIV) or admission to ICU in 32 patients.  Overt-DIC was diagnosed in 2 patients (6.2%), while 26 (81.2%) met the criteria for non-overt DIC and noted non-overt DIC score values >4 correlated with NIV/ICU and with thrombotic complications (P=0.04). A score >4 was the optimal cut-off value, performing better than the SIC score, while >4 with thrombotic complications were predictive   of   death. 

Overall Comments:

  1. The authors are to be congratulated for an interesting and robust data set despite the relatively smaller numbers of patients, the data is interesting and noteworthy. However, your manuscript has too many nonstandard abbreviations and, at times, is difficult to read even though English is my first language. Suggest that you spell out many of the nonstandard abbreviations such as pulmonary vein thrombosis (PVT), and a few others.
  2. Your table does not include fibrinogen levels; it was omitted. Please include this as this is quite important.
  3. In the introduction, please include either a table or some discussion of how the sepsis -induced coagulopathy and non-overt DIC scores are determined. This is not commonly known by many clinicians, including hematologists.
  4. Your discussion is relatively long and could be more tightly edited and shortened. By including some of the characteristics of SIC and DIC scoring in the introduction, it would be helpful as well as SOFA scoring, to understand how these scores are derived. I also think you need to better review the initial hypercoagulability of COVID 19 that can move to a more consumptive coagulopathic state, as mentioned later in my comments about your discussion.

Specific comments

Methods

1-Please change hematochemic two laboratory data or a more appropriate word.

2-Please note how the Sequential Organ Failure Assessment (SOFA) score is calculated

Laboratory values

  1. Please note fibrinogen levels are not listed. Please include
  2. You note the CTPA revealed the presence of PVT in 21 cases (65.6%). In all these patients, the dosage of
  3. The LMWH was adjusted to full anticoagulation. Please state what the doses of anticoagulation actually were adjusted to in terms of specific numbers
  4. You state “Non-overt DIC: *SE, SP, PPV and NPV……….” Spell these terms out or at least some of them as it is difficult to follow.

DISCUSSION

You state, “Alterations of the blood coagulation parameters are a common finding among COVID-19 patients. Elevated D-dimer levels are consistently reported, whereas their gradual increase during the disease course is associated with clinical worsening [11]. Furthermore, higher D-dimer and fibrin degradation products levels, as well as severe thrombocytopenia, elongation of PT and of aPTT, are typical of the severe stage of the disease, and have been associated with increased mortality in critically ill ICU patients, irrespectively of the presence of ARDS [12]. All these parameters are suggestive of life-threatening disseminated intravascular coagulation (DIC), which is likely triggered by complement-mediated microangiopathy following SARS-CoV-2 infection in its more aggressive shape [13, 14]”.  From this reviewer’s perspective, the pathophysiology is not quite as straightforward as you suggest and changes over time. The initial hypercoagulability that occurs in Covid 19 coagulopathy is associated with relatively normal prothrombin times and platelet counts but can evolve to a subsequent picture consistent with DIC and overt DIC, but remember that with time approximately 50% of patients are often co-infected with an additional bacterial another organism and develop a subsequent potential consumptive coagulopathy. The role of complement although may have an important role, is a multifaceted perspective, and other important things include increases in von Willebrand factor. Overall, the microangiopathy is clear, however multiple mechanisms are potentially responsible. I would be more descriptive and include an update of the recent perspectives on the microangiopathic lesions also, please note there also macro vascular lesion such as pulmonary emboli as reported by multiple investigators and up to 20% of cases including data by Helms, Klok, and others.  In this reviewer’s mind, the Tang and Chinese data is interesting, but we now have four more recent data that should be focused on.

Author Response

Response to Reviewer’s Comments #2

Dear Reviewer,

we are grateful for your preciuos time spent in the revision of our paper. Major and minor points have been addressed and changes have been made in the text. We believe that the paper has now been improved according to the Reviewer’s suggestion.

Hereafter You’ll find the detailed responses to your comments. Thanks again for your efforts.

The authors retrospectively reviewed their experience of non-overt coagulopathy in non-ICU patients with mild to moderate COVID-19 pneumonia to assess the occurrence of early coagulopathy and disseminated intravascular coagulation (DIC) in patients with mild to moderate respiratory distress from 18/03/2020 to 20/04/2020. Were retrospectively reviewed. They calculated both SIC and non-overt DIC scores and examined thrombotic complications, death, and worsening respiratory function requiring non-invasive ventilation (NIV) or admission to ICU in 32 patients.  Overt-DIC was diagnosed in 2 patients (6.2%), while 26 (81.2%) met the criteria for non-overt DIC and noted non-overt DIC score values >4 correlated with NIV/ICU and with thrombotic complications (P=0.04). A score >4 was the optimal cut-off value, performing better than the SIC score, while >4 with thrombotic complications were predictive   of   death.

Overall Comments:

  1. The authors are to be congratulated for an interesting and robust data set despite the relatively smaller numbers of patients, the data is interesting and noteworthy. However, your manuscript has too many nonstandard abbreviations and, at times, is difficult to read even though English is my first language. Suggest that you spell out many of the nonstandard abbreviations such as pulmonary vein thrombosis (PVT), and a few others.

We thank the Reviewer for his comment. A careful language revision has been performed by a native speaker and we hope that now the paper has been improved. Furthermore, a list of the abbreviations used has been added at the beginning of the paper (please see page 1-2, lines 38-61).

  1. Your table does not include fibrinogen levels; it was omitted. Please include this as this is quite important.

The correction has been performed in Table 2, as suggested (please see page 6). We apologize for the mistake.

  1. In the introduction, please include either a table or some discussion of how the sepsis -induced coagulopathy and non-overt DIC scores are determined. This is not commonly known by many clinicians, including hematologists.

As suggested by the Reviewer, the methods of determination of the SIC and non-overt DIC scoring systems have been moved from the discussion to the Methods section (we thought that it would have been more appropriate in this section rather than in the introduction, even if the main references of both scoring systems have been cited also in the introduction). Furthermore, Figure 1 has been added. Please see Methods on page 5, lines 140-147.

  1. Your discussion is relatively long and could be more tightly edited and shortened. By including some of the characteristics of SIC and DIC scoring in the introduction, it would be helpful as well as SOFA scoring, to understand how these scores are derived. I also think you need to better review the initial hypercoagulability of COVID 19 that can move to a more consumptive coagulopathic state, as mentioned later in my comments about your discussion.

As suggested by the Reviewer, the discussion was edited. The characteristics of SIC and DIC scoring have been moved to the Methods section and better explained using Figure 1 (see response #2). Furthermore, in the Methods we included a short paragraph and a new “Table 1” that better indicates how the SOFA score is calculated (Please see Methods page 3 lines 113-117 and page 4, line 124). Also, the pathogenetic mechanism that have been described for the coagulopathy of COVID-19 have been better elucidated according to the suggestion of the Reviewer and including the more recent findings of Helms and Klok (please see Discussion, page 9-10, lines 285-297  and references 14 and 15).

Specific comments

Methods

  1. Please change hematochemic two laboratory data or a more appropriate word.

The correction has been made as suggested on page 3 line 105.

  1. Please note how the Sequential Organ Failure Assessment (SOFA) score is calculated

In the Methods we included a short paragraph and a new “Table 1” that better indicates how the SOFA score is calculated (Please see Methods page 3 lines 113-117 and page 4, line 124).

Laboratory values

  1. Please note fibrinogen levels are not listed. Please include

The correction has been performed in Table 2, as suggested (please see page 6).

  1. You note the CTPA revealed the presence of PVT in 21 cases (65.6%). In all these patients, the dosage of the LMWH was adjusted to full anticoagulation. Please state what the doses of anticoagulation actually were adjusted to in terms of specific numbers.

The specification about the doses of anticoagulation were added as suggested by the Reviewer. Please see page 7 line 193.

  1. You state “Non-overt DIC: *SE, SP, PPV and NPV……….” Spell these terms out or at least some of them as it is difficult to follow.

 The terms were spelt out as suggested by the Reviewer (please see page 8 line 243-244). Furthermore, a list of abbreviation has been added at the beginning of the manuscript ((please see page 1-2, lines 38-61).

DISCUSSION

  1. You state, “Alterations of the blood coagulation parameters are a common finding among COVID-19 patients. Elevated D-dimer levels are consistently reported, whereas their gradual increase during the disease course is associated with clinical worsening [11]. Furthermore, higher D-dimer and fibrin degradation products levels, as well as severe thrombocytopenia, elongation of PT and of aPTT, are typical of the severe stage of the disease, and have been associated with increased mortality in critically ill ICU patients, irrespectively of the presence of ARDS [12]. All these parameters are suggestive of life-threatening disseminated intravascular coagulation (DIC), which is likely triggered by complement-mediated microangiopathy following SARS-CoV-2 infection in its more aggressive shape [13, 14]”. From this reviewer’s perspective, the pathophysiology is not quite as straightforward as you suggest and changes over time. The initial hypercoagulability that occurs in Covid 19 coagulopathy is associated with relatively normal prothrombin times and platelet counts but can evolve to a subsequent picture consistent with DIC and overt DIC, but remember that with time approximately 50% of patients are often co-infected with an additional bacterial another organism and develop a subsequent potential consumptive coagulopathy. The role of complement although may have an important role, is a multifaceted perspective, and other important things include increases in von Willebrand factor. Overall, the microangiopathy is clear, however multiple mechanisms are potentially responsible. I would be more descriptive and include an update of the recent perspectives on the microangiopathic lesions also, please note there also macro vascular lesion such as pulmonary emboli as reported by multiple investigators and up to 20% of cases including data by Helms, Klok, and others.  In this reviewer’s mind, the Tang and Chinese data is interesting, but we now have four more recent data that should be focused on.

We agree with the Reviewer’s perspective about the possible pathogenetic mechanisms of the coagulopathy of COVID-19. As suggested by the Reviewer, further explanations were added, taking in consideration the more recent findings of Helms and Klok (please see Discussion, page 9-10, lines 285-297  and references 14 and 15). The Discussion was therefore edited, and we believe that now has been qualitatively improved thanks to the suggestion.

Round 2

Reviewer 1 Report

Authors fully responded the comments. There is no further comment.

Reviewer 2 Report

Thank you for your thoughtful revision, well done.  I have no further comments.